# Long short-term memory (LSTM)-based news classification model

**Chen Liu** *

Nanjing Forestry University, Nanjing, Jiangsu, China

* njfuliuchen@163.com

## Abstract

In this study, we used unidirectional and bidirectional long short-term memory (LSTM) deep learning networks for Chinese news classification and characterized the effects of contextual information on text classification, achieving a high level of accuracy. A Chinese glossary was created using jieba—a word segmentation tool—stop-word removal, and word frequency analysis. Next, word2vec was used to map the processed words into word vectors, creating a convenient lookup table for word vectors that could be used as feature inputs for the LSTM model. A bidirectional LSTM (BiLSTM) network was used for feature extraction from word vectors to facilitate the transfer of information in both the backward and forward directions to the hidden layer. Subsequently, an LSTM network was used to perform feature integration on all the outputs of the BiLSTM network, with the output from the last layer of the LSTM being treated as the mapping of the text into a feature vector. The output feature vectors were then connected to a fully connected layer to construct a feature classifier using the integrated features, finally classifying the news articles. The hyperparameters of the model were optimized based on the loss between the true and predicted values using the adaptive moment estimation (Adam) optimizer. Additionally, multiple dropout layers were added to the model to reduce overfitting. As text classification models for Chinese news articles, the Bi-LSTM and unidirectional LSTM models obtained f1-scores of 94.15% and 93.16%, respectively, with the former outperforming the latter in terms of feature extraction.

**Data Availability Statement:** All relevant data are within the manuscript.

**Funding:** The author(s) received no specific funding for this work.

## Introduction

### Study background and significance

In this era of big data, the rapid progress of computational power and Internet technologies has led to the advent of 5G, which is profoundly influencing our lives. The Internet has become an indispensable part of everyday life, as we seek or transmit information daily. Consequently, the volume of data available on the Internet has grown exponentially [1]. Such data exist in many forms, including text, audio, and images. Textual information itself comes in many forms—for example, via blogs, forum posts, news, and email—and is particularly abundant. To manage this enormous volume of data, one must find methods to rapidly extract useful information from the text. This has important implications for policymakers' UX analysis, and decision making.

**Competing interests:** The author has declared that no competing interests exist.

Text classification is a common task in natural language processing (NLP) and has been studied extensively in academia because it can be used to manage enormous volumes of information. It is a common denominator in many innovative applications—such as sentiment analysis of product reviews, spam email screening, Weibo post-labeling, and intelligent recommendations [1].

Text classification is also important for news classification. News refers to information meant to keep the public informed of current events and social dynamics, and its content is predominantly presented in textual form [1]. Today, most newspaper publishers have transitioned to online publishing, which has improved the dissemination of news articles. For example, online news publishers make it possible for laymen to quickly obtain news related to topical events, such as the Covid-19 pandemic, Flight 5735 crash, and Ukraine–Russia war. However, the study of how to accurately locate news articles relevant to consumers' interests and needs remains worthwhile. Additionally, the screening and classification of news have important implications for the navigation of news websites and intelligent recommendation of their content.

Text classification models first emerged in the 1990s. These early models were based on classic machine learning algorithms—such as Bayesian classifier, k-nearest neighbor (KNN), and support vector machine (SVM) models—and used handcrafted features. Although these models have been successful to some extent, they all have limitations. For example, text is typically modelled as a bag of words, resulting in a loss of contextual information. This reduces the ability of a model to characterize textual features and can lead to excessively high dimensionality [1]. Additionally, most of these models rely on manual labeling, which greatly increases the classification effort and limits performance [1].

Over the past few years, the introduction of deep learning has culminated in major breakthroughs in speech, object detection, image identification, and facial recognition, gradually leading to advances in the field of NLP. Deep learning is highly proficient in extracting valuable information from existing textual information, which is impossible using traditional machine-learning algorithms [1]. Consequently, the advent of deep learning has ignited a new wave of interest in NLP, resulting in the emergence of highly sophisticated chatbots and translator bots. By eliminating the bottlenecks of traditional machine learning, it has become viable for NLP to be applied to real-world problems.

## Current state of text classification studies outside and inside China

The emergence of 5G has marked an era of accelerated growth in data volume. Textual data appears in various forms—including news articles, blog posts, and comments—at exceedingly high frequencies in our daily lives, accounting for a high percentage of the data generated on the Internet. Consequently, there is growing academic interest in the study of text classification.

**Current state of text classification studies outside China.** Text classification has been studied for a long time outside China, and many technical foundations have been laid by these studies. In the 1950s, text classification was performed manually. Typically, a few experts were tasked with creating text classification rules [2], which would then be used by workers to classify texts, an example of early expert systems. Although this was a viable method, it had a fatal flaw in that the rules were not reusable, making it necessary for experts to devote a large amount of time to rule revision. Consequently, the aim has been to find an efficient method to reduce such effort. For example, Luhn *et al.* [3] succeeded in developing a statistical method for encoding and searching text.

Machine learning algorithms were first used for text classification in the 1990s, with research focusing on two areas—namely, feature engineering and model optimization. Feature engineering is an important part of classification tasks. The most popular approach back then was the "bag-of-words" model, where one would define a "bag of words" and then record the frequency of each word. Although this method was simple and flexible, it overlooked the correlations between words, precluding the characterization of semantics or complexity. Typically, support vector machine (SVM), artificial neural network, and Bayesian classifier models have been commonly used for text classification.

In the 5G era, deep learning has become the most popular type of machine learning. Studies have shown that deep learning is far more adept at extracting information from texts than conventional machine-learning algorithms. Mikolov *et al.* published the *word2vec* tool to improve the efficiency of word vectorization and proposed various methods to optimize it [4]. Irsoy *et al.* found that recurrent neural networks (RNNs) could treat texts as sequences of words and capture their dependencies, which was impossible for many conventional machine-learning algorithms to do [5]. However, subsequent studies have found that RNNs cannot memorize data for a long time. The long short-term memory (LSTM) network was proposed to solve this problem, and it did so by having three gates to control the flow of information in and out of the memory cell. However, the conventional unidirectional LSTM model overlooked the context provided by the words that followed a target word. Consequently, the unidirectional LSTM model was extended to a bidirectional LSTM (BiLSTM) model to overcome this problem. The *seq2seq* model was created to handle multiple inputs and outputs. Subsequently, *seq2seq* models have been embedded in attention mechanism models, solving the tendency of these models to preserve information imperfectly during the feature extraction process.

**Current state of text classification studies in China.** Studies of NLP in China began in the 1980s. After Professor Hou reviewed the classic text classification methods and their technical details [6], many Chinese researchers began to study NLP techniques, leading to the emergence of various Chinese text classification methods based on the unique characteristics of Chinese text. Yin *et al.* [7] proposed the use of a short text classification algorithm based on convolutional neural network and KNN in order to solve the problem of high dimension and sparse data in the traditional K-Nearest Neighbor (KNN) short text classification algorithm based on TF-IDF.

Huan *et al.* [8] used the n-gram ensemble method to study real-time comment data and performed sentiment analysis on Internet comments. Wu *et al.* [9] used a dual-channel convolutional neural network (CNN) for text classification, which improved the extraction of textual features and thus the text classification performance. Ping *et al.* [10] improved the term frequency–inverse document frequency (TF–IDF) algorithm by accounting for the effects of the location and length of characteristic phrases on textual features. Jie *et al.* [11] combined *word2vec* with the TF–IDF method to develop a text classification method capable of assigning weights to word vectors.

Fan and Liu [12] introduced a method for feature extension that used concepts from Wikipedia to establish a set of semantic concepts to supplement semantic features in short Chinese texts, thus improving text classification performance. Yongjun *et al.* [13] proposed a method for expanding high-frequency words based on latent Dirichlet allocation, which reduced feature noise. They then used the TF–IDF method to convert short Chinese texts into vectors and compared their text classification method to other existing methods.

Ge *et al.* [14] proposed an LC-Transformer XL integration model for resolving the problems of sparse data and the inability to capture longer-distance dependencies between segments in text classification tasks.

The above review shows how text classification methods have evolved from manual operations to machine-learning-based tasks and now, deep-learning-based tasks. Several conclusions can be drawn from this analysis. First, deep learning greatly outperforms machine learning in text classification; second, the conversion of text into word vectors is a popular text-processing method; and third, LSTM-based algorithms are currently the most popular areas of research into text feature extraction.

## Proposed text classification method

Currently, most text classification algorithms use the same general procedure. First, a word vectorization tool is used to convert the text into word vectors. Next, a deep learning model is used to extract meaningful features from the text. Dimensionality reduction is then performed on these textual features, and the probability of each class of features is characterized. However, information loss inevitably occurs during the process.

To address this flaw, the proposed method uses a dual-layer LSTM structure in which the output of a BiLSTM network is passed to another LSTM network. This approach is informed by the latest developments in deep learning that have improved the integration of information and reduced information loss. This dual-layer structure reduces the loss of feature information, which improves the model accuracy and generalizability.

In this work, we focused primarily on text classification in the context of news articles. First, we used a web crawler to obtain approximately equal amounts of sports, entertainment, and technology news (data source: https://github.com/pengwei-iie/Bert-THUCNews). Next, the *jieba* tool was used to perform word processing on the data collected by the web crawlers. Stop-word processing was then performed on the segmented results. The datasets were divided into training (80%), testing (10%), and validation (10%) datasets. The *word2vec* tool was then used to convert the words into 200-dimension vectors, and a BiLSTM network was used to perform feature extraction. An LSTM network was then used to integrate the BiLSTM outputs. Finally, the integrated features were connected to the fully connected (FC) layer to obtain the probability of each feature class.

The novelty of this study can be expressed as follows:

1. **Feature extraction** is an important part of text classification. Therefore, a word vector model was trained to improve the word-to-word correlation and enhance the quality of the extracted features. This feature extraction strategy ultimately resulted in improved feature quality.

2. The dual-layer LSTM structure consists of both LSTM and BiLSTM models. This structure allows the results of the two models to be compared to examine whether contextual data have a major influence on text classification performance. In [1], the dual-layer LSTM network was optimized until it produced reasonably accurate classification results. The optimized model in [1] was employed as the basis for our method. Using two different LSTM models, this study demonstrated the use of an innovative method to solve text classification problems.

## Data processing and tasks

This section introduces the general ideas and task flow of the proposed task classification method and then describes the word segmentation, stop-word removal, and word vectorization processes.

## Task overview

News classification is a method of classifying enormous volumes of news texts. This is performed by first extracting words and text from news articles, and then converting the text into machine-readable mathematical features. These features are then used to construct classifiers for the classification task. In this text classification problem, it is necessary to first determine the class of each news text—thus, this is a supervised learning method.

The original objective of deep learning was to allow machines to perform calculations using humanlike neurons over a wide range of areas. In recent years, deep learning-based NLP has made tremendous strides and is now considered to be a relatively mature field of study. In general, text classification requires data preprocessing, text representation, classifier construction, and performance evaluation. This process flow is shown in Fig 1.

Text representation has been shown to be a crucial part of this process. If the text representation is poor, it is impossible to achieve high accuracy, even with an outstanding classifier algorithm. Conversely, if text representation is good and the features of each class are clearly distinct, even a simple classifier can provide adequate levels of accuracy. In other words, the lower and upper boundaries of the performance of a text classification model are determined by the quality of its text representation and the complexity of its classifier, respectively.

Because the feature extraction process and classifier of the BiLSTM model are combined in the proposed method, the performance of the model depends on the quality of feature extraction, which is also a part of the classification process. Consequently, it is evident that every step of the text classification process is important.

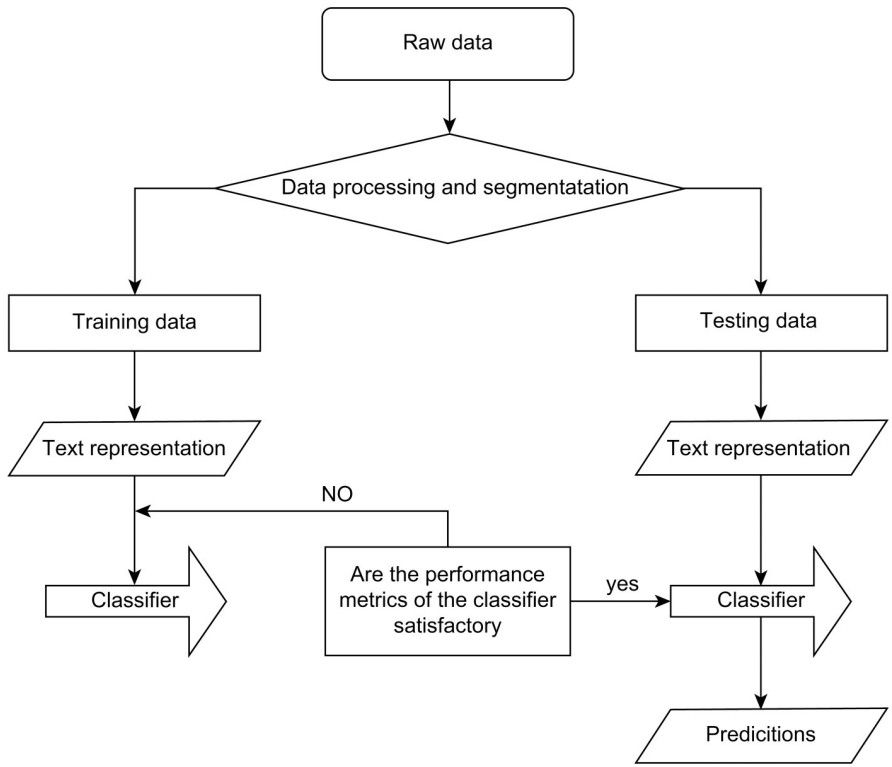

**Fig 1. NLP classification process flow.**

## Processing of textual data

The experimental textual datasets used in this study were obtained using a web crawler. However, these data were poorly organized and included non-textual data (such as images and links). Therefore, the data had to be preprocessed before word segmentation to remove all non-news data. This process included data deduplication, blank removal, and deletion of images, links, and non-Chinese characters. In this study, regular expressions were used to obtain Chinese news articles from the URLs in the dataset.

Because the physical layer of a computer can only recognize digital data, it was necessary to include a preprocessing step in the text classification model. First, all symbols and meaningless characters in the text were removed. The processed data were then converted into digital data that could be recognized by a computer, which was a prerequisite for all subsequent text processing procedures.

(1) *Word segmentation*: Word segmentation refers to the segmentation of a sentence into individual words by using a suitable algorithm. Because Chinese words are more difficult to segment than English words, each sentence is segmented into sequences consisting of individual characters. For example, the segmentation of "上赛季塞尔塔在主帅库代的带领下最终杀进了西甲前八强, 并替代杯赛冠军参加了欧罗巴战场" would result in: "上赛季/塞尔塔/在/主帅/库代/的/带领/下/最终/杀进/了/西甲/前/八强/, /并/替代/杯赛/冠军/参加/了/欧罗巴/战场."

Currently, the three most commonly used word-segmentation algorithms are string-matching, probability-based segmentation, and semantic segmentation. String matching is performed by constructing a dictionary, matching words in the text to the dictionary, and then performing word processing for successful matches. However, a disadvantage of this method is that it cannot recognize new words [15]. Probability-based segmentation is performed by evaluating the probability of the co-occurrence between one character and its adjacent character, and then considering the characters as a word if the probability passes a certain threshold. The disadvantage of this method is that it requires a corpus, which requires an enormous amount of time and effort. Semantic segmentation is rarely used because it is still immature. The most commonly used Chinese word-segmentation tools are HanLP, THULAC, and *jieba*. In this study, *jieba* was used to perform word segmentation on news data captured by a web crawler.

(2) *Stop-word removal*: Texts often contain redundant words. Although these words do not greatly affect the efficacy of text classification, they can result in high data dimensionality. Therefore, stop-word removal must be performed on word-segmented data. It refers to the removal of punctuation marks and frequently used words that do not contribute to classification of the segmented data, as they have no meaning and only play grammatical roles. Stop words in Chinese include prepositions and conjunctions like "的," "了," and "呢." The removal of meaningless words simplifies the text while retaining the words that are more important for classification.

## Text representation

After the textual data have been preprocessed, they must be converted into binary data because a computer cannot handle unstructured human language. Text representation refers to the use of a set of numbers to represent the core information contained in a text. However, the type of information preserved during this process differs widely from one method to another, and it is crucial to preserve as much information as possible during text representation. Consequently, it is important to select an appropriate text-representation method.

One of the most commonly used methods for text representation is the bag-of-words model [16], which is simple and flexible. In this approach, a bag of words is first defined and each word in the bag is recorded along with its frequency. However, the sequence in which the words appear or word-to-word correlations are not recorded, leading to the loss of contextual information [16].

With deep learning, each word can be represented as a word embedding—that is, a distributed representation in a neural network. The strength of this approach is that all words can be represented in a low-dimensional vector space, allowing the relationship between a pair of words to be characterized using the distance between their vectors. Word embedding is an excellent method of preserving word-to-word relationships and the *word2vec* tool from Google is currently the most commonly used word vectorization tool.

### Word vectorization models

**WORD2VEC.** *word2vec* is a word vectorization tool published by Google and is based on conventional neural network models trained to convert words into word vectors. The models used in *word2vec* are the continuous bag of words (CBOW) and skip-gram models, with the latter being the inverse of the CBOW model [17]. *word2vec* is a computationally efficient model and is easy to use because it can be trained using a large word-segmented corpus to produce a table of word vectors. The vector of each word can then be obtained by reviewing the table.

**CBOW.** The CBOW model conditionally predicts a target word based on the words that come before and after it—that is, its context. It is an extension of early neural network language models that only consider words before the target word.

Furthermore, it moves away from the second layer of these language models [17], making it a three-layer network. Fig 2 shows the architecture of the CBOW using two words that appear before and after the target word as an example.

In Fig 2, $w(t)$ is the target word, and $w(t−1)$, $w(t−2)$, $w(t+1)$, and $w(t+2)$ are the contexts of the target word; they are all one-hot encoded vectors [17]. The result is passed from the input layer to the hidden layer, as follows:

$$V^1 = \frac{1}{2 \cdot 2} \sum_{i=1}^{2 \cdot 2} W_1 * w(i) \tag{1}$$

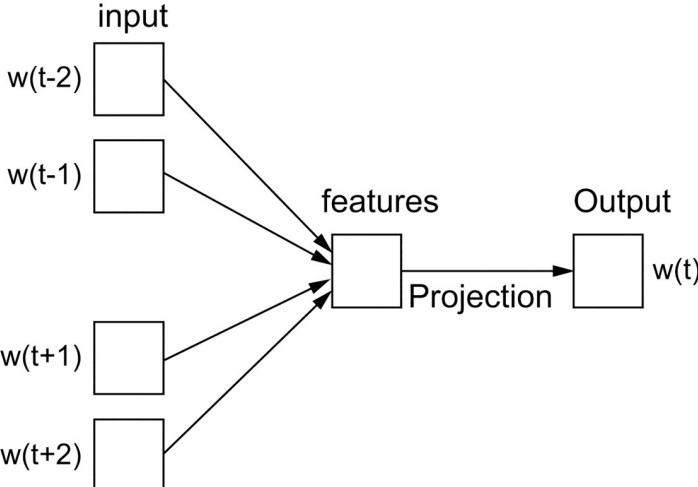

**Fig 2. Architecture of the CBOW model.**

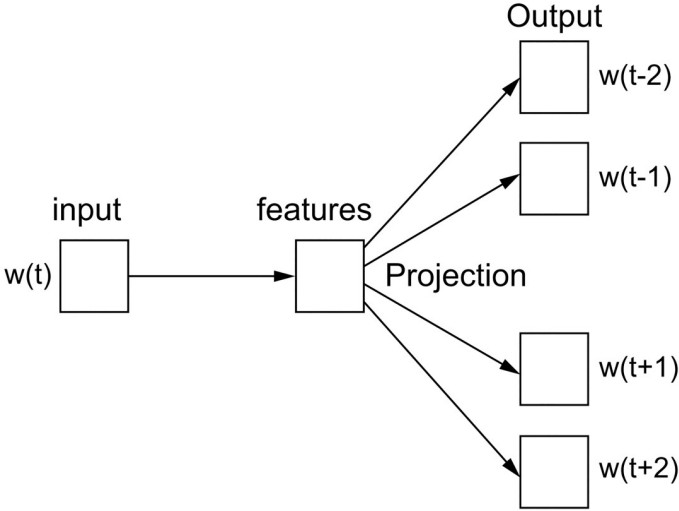

**Fig 3. Architecture of the skip-gram model.**

where W1 denotes the weight of the connection from the input layer to the hidden layer, 2·2 refers to the number of words in the input (i.e., the two words before and after the target word).

Hidden layer -> output layer:

$$y = softmax\left(W_2 V^I\right) \tag{2}$$

where $W_2$ denotes the weight of the connection from the hidden layer to the output layer, *softmax* denotes the activation function, and $V^I$ denotes the vector representation of $w(t)$. This produces the predicted probability of each word, calculated loss of $w(t)$, and optimized weights [17].

**Skip-gram model.** The skip-gram model is the exact inverse of the CBOW model in that it uses an input word to predict the surrounding words (context). This model is based on the same principles as the CBOW model. Its architecture is shown in Fig 3.

## Theory: Principles and explanation of the proposed model

There are many text classification algorithms available today, which can be broadly divided into training datasets and word-segmentation-based algorithms. Moreover, the type of learning can be supervised, semi-supervised, or unsupervised. Algorithms that are broadly suitable for text classification include deep learning, support vector machine (SVM), k-nearest neighbor (KNN), and Bayesian classifiers [18]. In this study, the LSTM and BiLSTM models were used for news classification, and their training was performed using the adaptive moment estimation (Adam) optimizer with a dropout layer. This section describes the architecture of the proposed model.

### Activation function

Neural networks generally require activation functions as they are necessary for fitting nonlinear relationships. Here, we describe the activation functions used in the proposed method and the scenarios in which they can be applied. The activation function used in this method can be

expressed as follows:

$$sigmoid = \frac{1}{1 + e^{-x}} \tag{3}$$

The *sigmoid* function has an output range of [0, 1]. In other words, it normalizes all inputs by compressing them to the [0, 1] range, and its output is typically used as the predicted probability during binary classification.

$$tanh(x) = \frac{1 - e^{-2x}}{1 + e^{2x}} \tag{4}$$

The output of the *tanh* function (hyperbolic tangent function) is wider than that of *sigmoid* function at [−1, 1], as it converges more quickly and is less susceptible to vanishing gradients.

$$softmax : p(x = i) = \frac{e^{z_i}}{\sum_{c=1}^{c=C} e^{z_c}} \tag{5}$$

where $z_i$ denotes the $i^{th}$ input, $p(x = i)$ denotes the predicted result, and $C$ denotes the dimensionality of the input. *Softmax* also compresses all inputs into the [0, 1] range, and its output sums to one. It is typically used in the last layer of a classification neural network, and can be used to represent the probability of each type.

## LSTM

LSTM networks are a type of recurrent network aimed at addressing the vanishing gradient problem in traditional RNNs, thereby improving their inability to store long-term memories. As a result, LSTM networks have a tremendous capacity for remembering information, and are currently the most commonly used models for NLP. The LSTM network has three gates—that is, the forget, input, and output gates. This solves the key limitation of RNNs—hat is, their short memory [19]. These gates determine the bits of information that enter and leave the memory cells or refresh the network memory. Its architecture is shown in Fig 4.

Here, $C_{t-1}$ is the previously remembered information of the LSTM network, $h_{t-1}$ is the previous LSTM output, and $x_t$ is the current input. $C_t$ is the remembered information from the current input, $h_t$ is the output of this layer, and $\sigma$ is the activation function.

The architectures of the LSTM gates are shown below: [20].

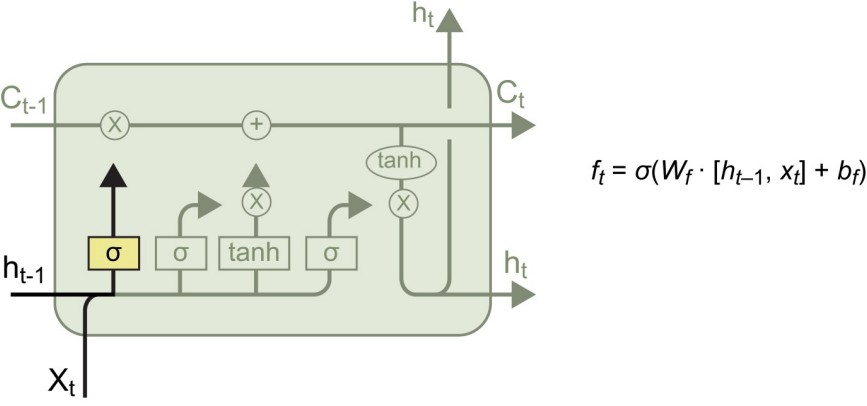

$$f_t = \sigma(W_f \cdot [h_{t-1}, x_t] + b_f)$$

**Fig 4. Architecture of the LSTM network.**

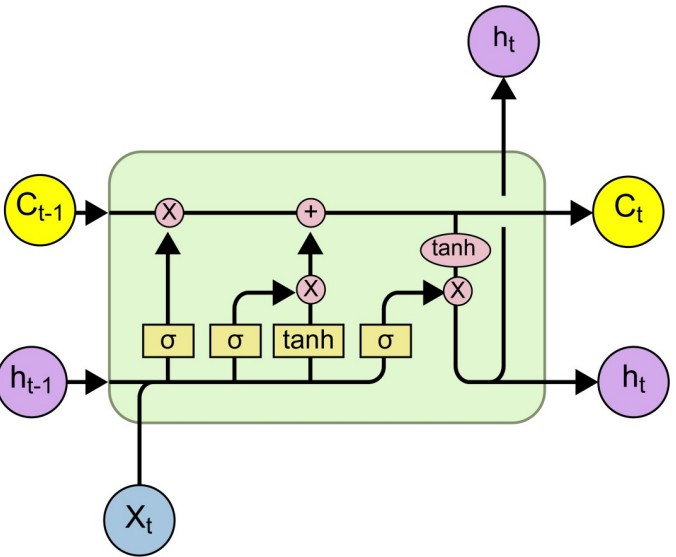

**Fig 5. The forget gate.**

**Forget gate.** This gate determines the ratio of information to be forgotten, its structure being as shown in Fig 5.

Here, $\sigma$ is the sigmod function, which compresses $f_t$ to [0, 1]. The product of $f_t$ and $C_{t-1}$ determines the information that is retained in $C_{t-1}$. $[h_{t-1}, x_r]$ indicates that $h_{t-1}$ is concatenated with $x_t$, and $W_f$ and $b_f$ are the weight and bias, respectively, which must be optimized.

**Input gate.** This gate determines the information to be remembered in $x_t$ and $h_{t-1}$, its structure being as shown in Fig 6.

As shown in Fig 7, the input gate consists of two parts. The first part is the *sigmoid* function, which has the same role as the forget gate—that is, to compress $i_t$ to [0, 1] and determine the amount of information to be remembered in ~Ct. The second part is the *tanh* function, which integrates the information from $x_t$ and $h_{t-1}$ and transforms the data into the range [−1, 1], thus constructing a vector of new candidate values ~Ct that can be added to the state. Thus, the

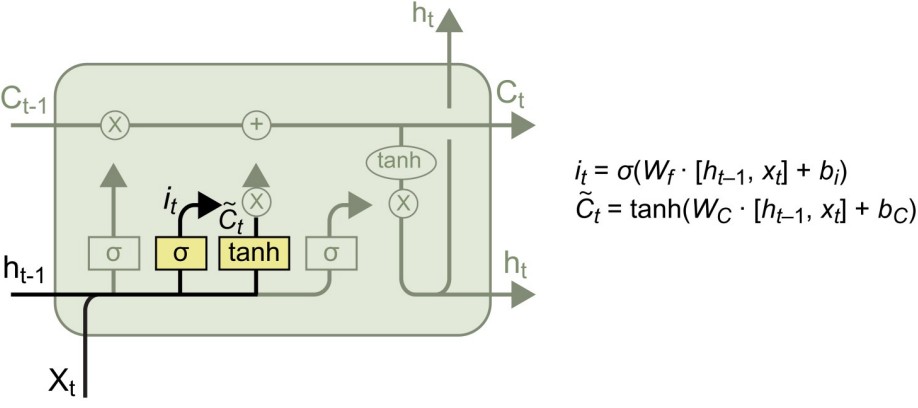

$$i_t = \sigma(W_f \cdot [h_{t-1}, x_t] + b_i)$$
$$\tilde{C}_t = \tanh(W_C \cdot [h_{t-1}, x_t] + b_C)$$

**Fig 6. The input gate.**

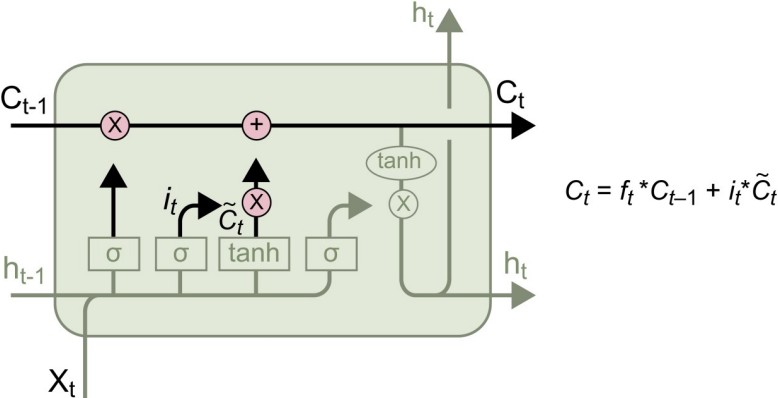

**Fig 7. Updating the cell state.**

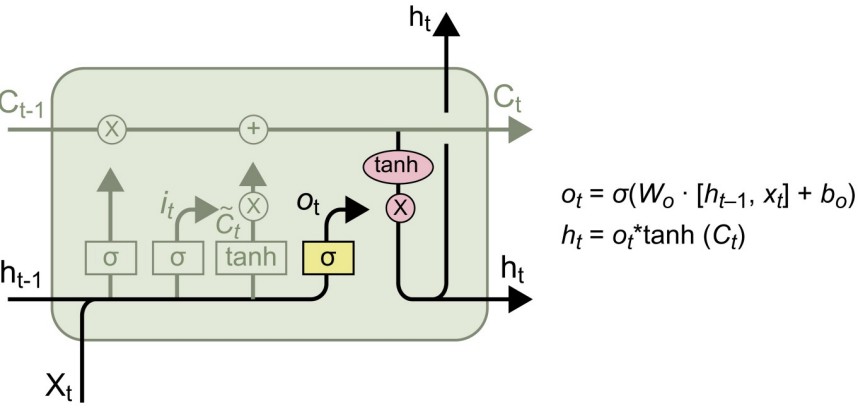

**Fig 8. The output gate.**

outputs of the input gate are $i_t$ and ~Ct. The update of the old cell state $C_{t-1}$ to the new cell state Ct is illustrated below:

**Output gate.** Here, the output of the forget gate, $f_t$, is multiplied by $C_{t-1}$ to determine the information that will be retained in the next cell. $i_t$ is then multiplied by ~Ct to determine the information to be remembered from the input. Finally, this information is added to update the cell state to $C_t$. The structure of the output gate is shown in Fig 8.

Here, $\sigma$ is the *sigmod* function and $h_t$ is the output of this layer. First, the *sigmoid* structure is used to obtain $o_t$ and calculate the amount of information to be output. $o_t$ is then multiplied by $tanh(C_t)$ to determine the output [21].

## BiLSTM

BiLSTM is an extension of the LSTM model and consists of two LSTMs running in two directions. The original LSTM can only consider information that precedes the current word, which may not be sufficient in certain cases.

Therefore, the BiLSTM model was developed to consider context in both directions. The architecture of the BiLSTM model is shown in Fig 9.

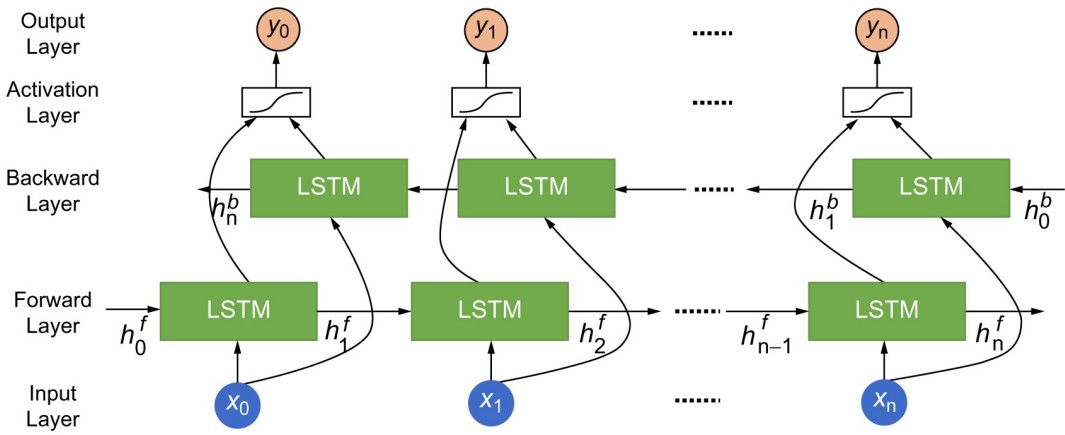

**Fig 9. Architecture of the BiLSTM model.**

## Fully connected layer

After feature extraction is performed using the BiLSTM network, the output from each layer is stored. Information integration is then performed in the LSTM layer, and the final output ($h_t$,) is retained as a feature vector of the text.

This feature vector is then used to construct a fully connected layer between the BiLSTM and LSTM networks to obtain the probability of each class. The architecture of this network is illustrated in Fig 10. Here, $X_n$ is the input, and its propagation from the input layer to the hidden layer can be expressed as follows:

$$hide1 = \sigma(whX + bn) \tag{6}$$

where $\sigma$ denotes the *tanh* function, $w_h$ denotes the weight matrix of $X$, $b_h$ denotes the bias and $hide_1$ denotes the hidden layer. Eq (6) shows the propagation of the LSTM output features to the hidden layer. Similarly, the structure of the hidden layer can be used to obtain the probability of each class, as follows:

$$Oj = \sigma(wh + 1 * hidel + bh + 1) \tag{7}$$

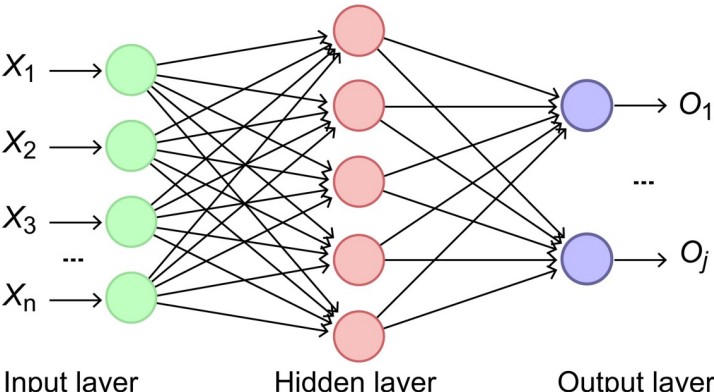

**Fig 10. Structure of the fully connected layer.**

where $\sigma$ denotes the *softmax* function, $o_j$ denotes the probability of class $j$, $w_{h+1}$ denotes the weight matrix of $hide_l$ and $b_{h+1}$ denotes the bias of the output layer. Thus, the LSTM-extracted features are passed through a fully connected layer to obtain the probability of each class.

## Loss function

The loss function measures the difference between the predicted and true values and usually outputs a positive value. In general, the lower the loss, the better the performance of the model.

During the training phase, the input data undergoes forward propagation, and the difference between the predicted and true values is calculated using the loss function to obtain the loss. After the loss is determined, a gradient descent algorithm can be used to reduce it. This gradually brings the model predictions closer to the true values, thereby achieving the goal of model learning [3].

The loss function used in the proposed method is categorical cross-entropy, which can be expressed as follows:

$$Loss = -\sum_{i=1}^{C} y_i \cdot log(\hat{y}) \tag{8}$$

where $y_i$ denotes the true value, $\hat{y}_I$ denotes the predicted value, $C$ denotes the number of classes, and $y_i$ denotes the one-hot encoding of the categorical data. In this model, the loss is reduced to make $y_i$ and $\hat{y}_I$ as similar as possible.

## Gradient descent algorithm

In this study, the Adam optimizer was used to optimize the hyperparameters of the deep learning network [22]. Adam optimization comprises five steps, as follows [3]:

Step 1: Estimate the first-order moment. Here, $m_0 = 0$, and $\beta_1$ is the rate of decay, which is 0.9 by default.

$$m_t = \beta_1 m_{t-1} + (1 - \beta_1)g_t \tag{9}$$

Step 2: The second-order moment is estimated. Here, v0 = 0, β2 is the rate of decay of the estimated second-order moment, which is 0.999 by default.

$$v_t = \beta_2 v_{t-1} + (1 - \beta_2)g_t^2 \tag{10}$$

Step 3: Correct the bias of the estimated first-order moment.

$$\hat{m}t = \frac{m_t}{1 - \beta_1^t} \tag{11}$$

Step 4: Correct the bias of the estimated second-order moment.

$$\hat{v}t = \frac{v_t}{1 - \beta_2^t} \tag{12}$$

Step 5: Update the hyperparameters of the model Here, $\alpha$ denotes the learning rate, $\varepsilon$ denotes a constant $(10^{-8})$ that prevents the denominator from going to 0, $\theta$ denotes the parameter that is being updated and optimized, and $g_t$ denotes the gradient, which is a vector consisting of the biases in each dimension.

$$\theta_t = \theta_{t-1} - \alpha * \hat{m}_t / \left( \sqrt{\hat{v}_t} + \varepsilon \right) \tag{13}$$

It is evident that the Adam algorithm controls the exponential moving average of gradients and square gradients, but not the gradient itself [19].

## Batch size

For a small dataset, adequate performance can be obtained without setting the batch size. On large datasets, it is not realistic to input all of the data in a single pass, as this can lead to an "out of memory" (OOM) exception. Therefore, the data must be divided into batches when training is performed using large datasets, the "batch size" corresponding to the number of samples processed in each training instance [23]. The reasons for defining a batch size can be summarized as follows:

1. The model can be trained on datasets of arbitrary sizes without OOM exceptions.

2. The number of epochs can be reduced as a larger number of epochs is required to achieve the same precision if all data are input into one training instance.

The batch size should be selected carefully. Although a large batch size improves the gradient accuracy, it also increases the training time and can cause an OOM exception. Conversely, a small batch size can make it difficult for the model to converge, reducing model accuracy.

## Dropout

In deep learning, overfitting can occur if the training dataset is small, the number of parameters being trained is large, and the number of epochs is large. Overfitting may be intuitively understood as poor generalizability; the model may perform well on the training dataset, but it will perform poorly on the testing dataset. The most commonly used methods to prevent overfitting are early stopping and dropout regularization. The proposed method uses dropout regularization to prevent overfitting. Dropout causes some neurons to be randomly ignored during training, thereby preventing an overreliance on the information contained by these neurons. This improves generalizability and greatly reduces overfitting [21].

## Methodology: Model design and training

The previous section introduced the LSTM and BiLSTM models and several concepts related to model optimization. In this section, these models are optimized within an experimental environment until adequate optimal solutions are produced.

## Experimental environment

The models were constructed using news data obtained from a web crawler, 80% of the data being used for the training dataset, 10% being used for the validation dataset (to adjust the models' hyperparameters during model training), and 10% being used for the testing dataset (to evaluate the model accuracy). The experimental setup is summarized in Table 1.

**Table 1. The experimental environment and software tools.**

| Item | Description |
|---|---|
| Operating system | Windows 10 |
| Random access memory | 16 GB |
| Open-source framework | Tensorflow2.8.1, Keras2.4.3 |
| Programming language | Python3.10 |
| Development environment | Jupyter Notebook |
| Word segmentation tool | *jieba* |
| CPU | IntelCore i7-10750H CPU @ 2.60 GHz |

The model was developed on a Jupyter Notebook using the Python 3.10 programming language and Keras 2.4.3 deep learning API in the Anaconda development environment. The operating system used was Windows 10, running on a 2.60-GHz Intel Core i7-10750H CPU.

## Experimental data

News articles on sports, entertainment, and technology were collected using a web crawler that we coded ourselves, the results being stored in three separate datasets. These datasets contained 2,491, 2,360, and 2,296 texts on entertainment, sports, and technology, respectively, for a total of 7,147 texts. The number of texts in the validation, testing, and training datasets were 715, 715 and 5,717, respectively. The word-count distribution of the processed textual data is shown in Fig 11.

As shown in Fig 11, approximately 5,000 texts have word counts less than 200. Therefore, only the first 200 words were used in this study. There were two other reasons for this choice: 1) Computational constraints: it took a long time to train the model using just 200 words from each text. 2) Accuracy: with 50 words from each text, the model achieved an accuracy of 97% with the training dataset, but only ~70% with the testing dataset, which is indicative of severe overfitting. This was greatly improved by selecting 200 words from each text.

## Data processing flow of the model

To process the enormous volumes of Chinese text required to intelligently classify news articles [18], a BiLSTM-based text classification model was used in this study. The data processing flow of which is shown in Fig 12.

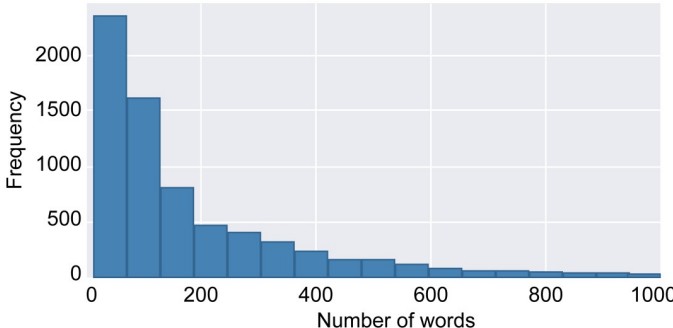

**Fig 11. Histogram of word counts.**

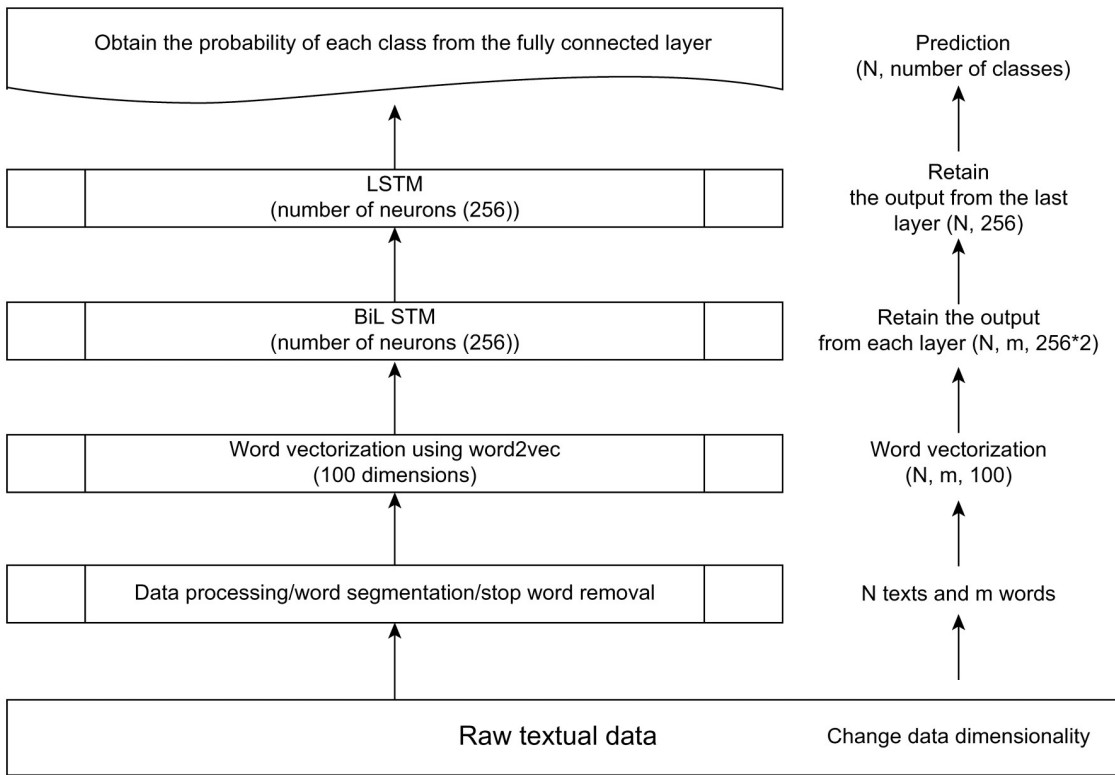

**Fig 12. Data processing flow of the proposed model.**

Step 1: Because raw data contain large volumes of redundant information, it is necessary to first perform word segmentation, stop-word removal, and low-frequency word filtering to preserve only the most valuable information in the textual data.

Step 2: Before a deep-learning algorithm can be used to process textual data, it must be converted into a more efficient matrix. Therefore, in Step 2, the preprocessed data from Step 1 are converted into matrix form. As the texts have different word counts, texts with fewer than 200 words are filled with blank strings, whereas texts with more than 200 words are simply cropped to the first 200 words. The *word2vec* tool can be used to convert these words into word vectors, thus transforming the texts into matrices of equal dimensionality.

Step 3: The BiLSTM network is used to extract contextual features, and the output from each layer of the network is collected.

Step 4: The LSTM network is used to integrate the outputs of the BiLSTM network, and only the output of the last layer is returned. This output is then treated as integrated information.

Step 5: The integrated information is connected to a fully connected layer to obtain the probability for each class.

## Setting of configurable parameters

The same model configuration (Table 2) was used in all the experiments. The first layers of the LSTM and BiLSTM networks contained the same number of neurons. The configurable parameters of the models were as shown in Table 2.

**Table 2. Table of configurable parameters.**

| Parameter | Values |
|---|---|
| $\alpha$ | 0.0005 |
| Batch Size | 32 |
| Word vector dimensionality | 100 |
| Dropout | 0.8 |
| Epochs | 25 |
| Number of neurons in the BiLSTM network | 256 |
| Number of neurons in the LSTM network | 256 |
| Number of neurons in the hidden layer of the fully connected layer | 100 |

Here, $\alpha$ denotes the learning rate of the gradient descent algorithm, 'epochs' denotes the number of training instances needed to go through all of the data, 'batch size' denotes the quantity of data used in each training instance, 'word vector dimensionality' denotes the dimensionality of the word vector generated by the *word2vec* tool, and 'dropout' denotes the ratio of neurons that are randomly ignored during each training instance. The BiLSTM and LSTM networks have the same number of neurons, and the 'number of neurons in the LSTM' is the number of LSTM neurons used for information integration.

## Performance metrics

Many metrics can be used to measure the performance of a classification model, including *accuracy*, *recall*, *precision*, and $F_1$-score (The $F_1$-score synthesizes the precision and recall scores, which are the harmonic means of precision and recall. In the formula, β usually takes 0.5,1,2 to represent the weight ratio of accuracy. In this paper, β takes 1 for the calculation). The equations used to calculate these performance metrics can be expressed as follows in Table 3:

$$precision = \frac{TP}{TP + FP} \tag{14}$$

$$recall = \frac{TP}{TP + FN} \tag{15}$$

$$F1 = \frac{2 * precision * recall}{recall + precision} \tag{16}$$

$$accuracy = \frac{TP + TN}{TP + TN + FN + FP} \tag{17}$$

**Table 3. Performance metrics.**

|  | True positive | True negative |
|---|---|---|
| Predicted correctly | TP | FP |
| Predicted wrongly | FN | TN |

*Precision* and *recall* are usually negatively correlated, as high *precision* values are usually associated with low *recall* values and vice versa. The $f_1$-score is a combined measure that takes the harmonic mean of *precision* and *recall*, and is the most commonly used metric for classification models [24].

## Results: News article classification

### Individual classification performance of the BiLSTM and LSTM networks

Two experiments were designed to examine the effects of contextual information on classification performance. In the first experiment, text classification was performed using the unidirectional LSTM network alone. In the second experiment, text classification was performed using the BiLSTM network.

These experiments were performed in the same experimental environment to facilitate comparison. The results obtained using the LSTM network are listed in Table 4.

In Table 4, it is evident that the *recall* decreases with increasing *precision*, and the $f_1$-scores of the network for sports, entertainment and technology news are 0.9444, 0.9280, and 0.9224, respectively. Thus, the unidirectional LSTM network performs well in the extraction of textual information. Moreover, as these classes of text are quite different from each other and the use of *word2vec* facilitates the preservation of contextual information, it is relatively easy to distinguish them using LSTM-extracted features.

The results of the bidirectional BiLSTM network are shown in Table 5.

In Tables 4 and 5, *precision* and *recall* are negatively correlated. Thus, the classification performance is usually evaluated using the $f_1$-score instead, the $f_1$-scores of the BiLSTM network for sports, entertainment, and technology news being 0.9448, 0.9607, and 0.9190, respectively.

We could then evaluate the classification performance of the BiLSTM and LSTM networks using the overall *accuracy*, *recall* and $f_1$-score, the results of which are shown in Table 6.

Table 6 shows that both models are able to achieve accuracies greater than 90% after the words are vectorized using *word2vec*. Furthermore, the BiLSTM network outperforms the unidirectional LSTM network.

To delve deeper into the differences between these networks, the *loss* vs. *epoch* and *accuracy* vs. *epoch* curves of these models on the training and validation datasets were plotted, as shown in Fig 13.

It is evident that the BiLSTM network stabilizes and converges more quickly than the unidirectional LSTM network. The BiLSTM network is better able to extract contextual features

**Table 4. Classification performance of the LSTM network.**

| LSTM | Precision | Accuracy | Recall | $f_1$-score | Number of texts |
|---|---|---|---|---|---|
| Sports | 0.9526 | 0.9426 | 0.9364 | 0.9444 | 236 |
| Entertainment | 0.9865 | 0.9634 | 0.8760 | 0.9280 | 250 |
| Technology | 0.8659 | 0.8493 | 0.9869 | 0.9224 | 229 |

**Table 5. Classification performance of the BiLSTM network.**

| BiLSTM | Precision | Accuracy | Recall | $f_1$-score | Number of texts |
|---|---|---|---|---|---|
| Sports | 0.9862 | 0.9761 | 0.9068 | 0.9448 | 236 |
| Entertainment | 0.9957 | 0.9801 | 0.9280 | 0.9607 | 250 |
| Technology | 0.8566 | 0.8512 | 0.9913 | 0.9190 | 229 |

**Table 6. Overall classification performance.**

| Model | Accuracy | Recall | $f_1$-score |
|---|---|---|---|
| BiLSTM | 0.9413 | 0.9420 | 0.9415 |
| LSTM | 0.9315 | 0.9331 | 0.9316 |

that are useful for classification, which improves the accuracy and reduces the loss relative to the unidirectional LSTM network.

The confusion matrices of these two models are shown in Fig 14.

It is evident that BiLSTM network is more accurate than the LSTM network in the classification of technology and entertainment news but less accurate for sports news, which may be caused by overfitting in the BiLSTM network.

## Individual classification performance of the BiLSTM network and other common models

Several commonly used deep learning models were chosen for the comparison experiments, namely the FastText, BERT, and TextCNN models. Different degrees of parameter

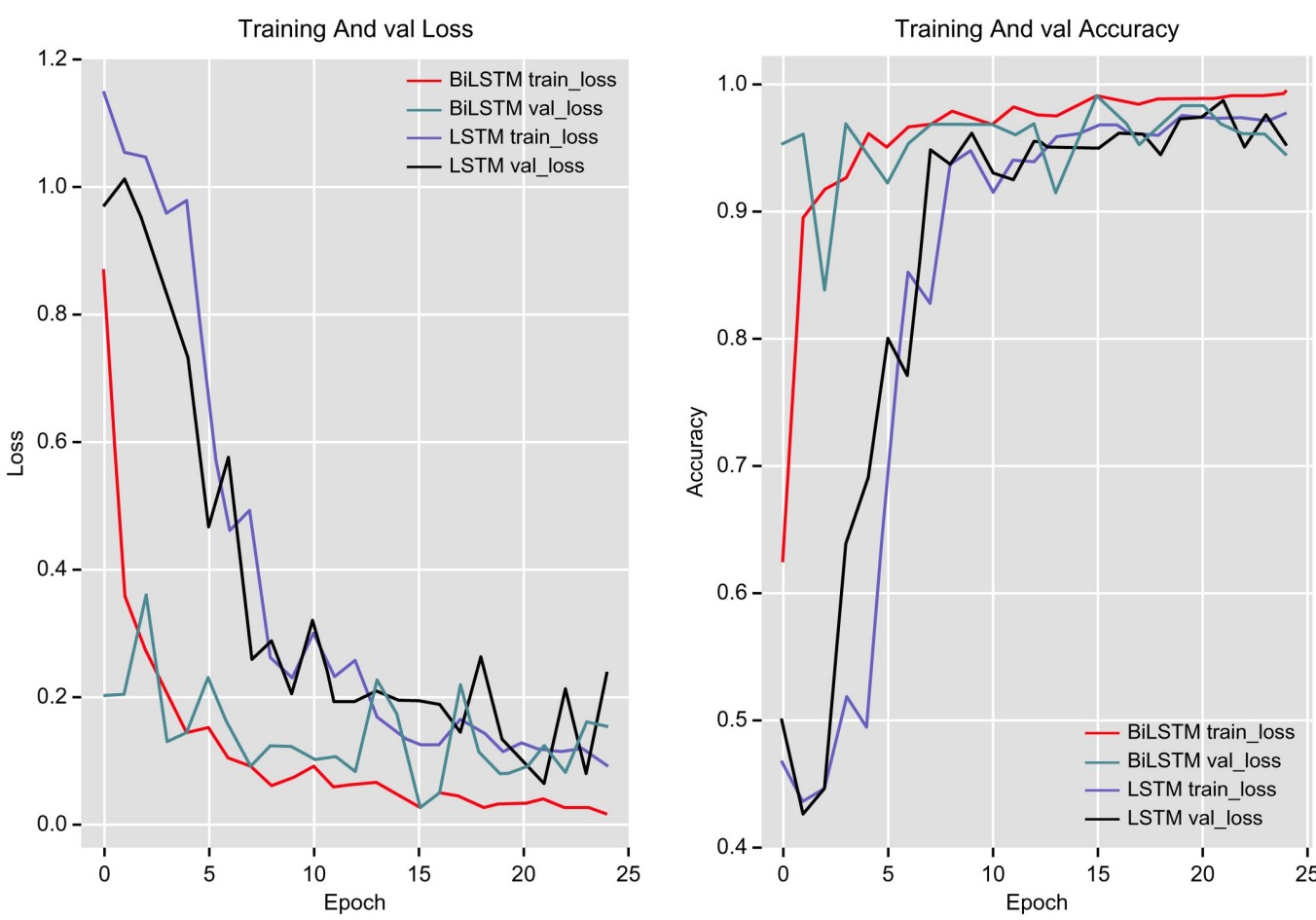

**Fig 13. Loss vs. epoch and accuracy vs. epoch on the training and validation sets.**

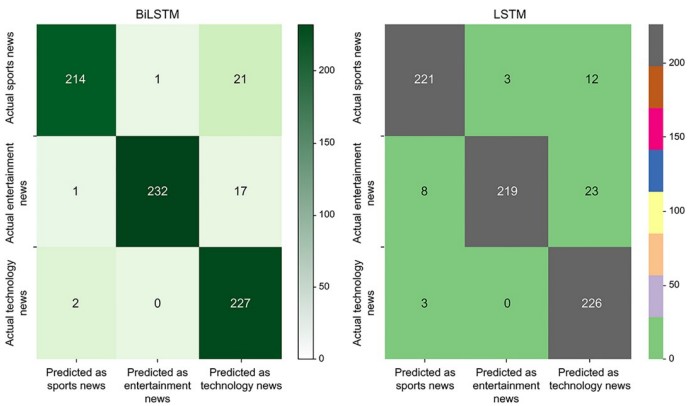

**Fig 14. Confusion matrices of the BiLSTM and LSTM networks.**

adjustments were made for the above models according to the situation, so that the models can perform better for the comparison of experimental results. The results of the comparison experiments on the test set are shown in Table 7.

From the results, it can be seen that the BiLSTM network has a higher accuracy rate and better model performance compared to the other benchmark models. In order to further compare the experimental results of each model in a more intuitive way, this experiment plots the statistics of the results of each model, as shown in Fig 15.

Overall, the BiLSTM network, due to its internal Transformer structure and its unique training method, and built on pre-training, has a very strong ability to learn the correlation between contexts, and therefore gives the best classification results.

## Section summary

In this study, the *word2vec* tool was used to map the preprocessed words into word vectors. The LSTM-based BiLSTM architecture was then used to extract contextual information from these word vectors. The experimental results showed that the BiLSTM network outperformed the unidirectional LSTM architecture and achieved an overall accuracy of 94.13%. Consequently, it can be concluded that contextual information plays an important role in text classification. In addition, performance comparisons also show that the bilstm network works better than other common models such as BERT.

## Conclusions

NLP is a major component of artificial intelligence, and there is currently a great deal of interest in technologies such as chatbots and customer service bots. Text classification has always been a core aspect of NLP. The Internet is constantly generating enormous volumes of news-

**Table 7. Classification performance of the model comparison experimental.**

| Model | Precision | Accuracy | Recall | f1-score |
|---|---|---|---|---|
| FastText | 0.9062 | 0.8994 | 0.9028 | 0.9058 |
| BERT | 0.9457 | 0.9379 | 0.9391 | 0.9313 |
| TextCNN | 0.9211 | 0.9162 | 0.9186 | 0.9197 |
| BiLSTM | 0.9631 | 0.9514 | 0.9574 | 0.9544 |

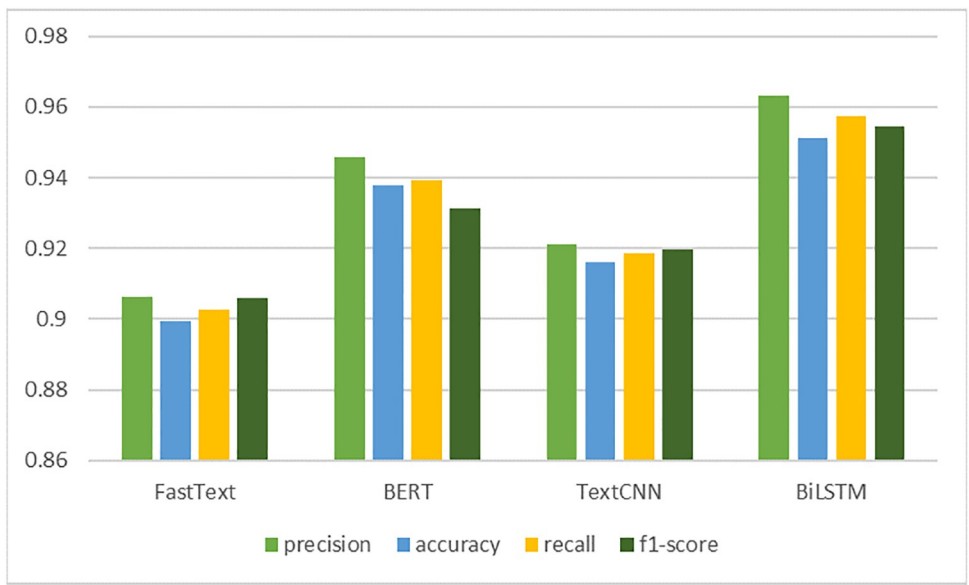

**Fig 15. Statistical chart of the results of the model comparison experiment.**

related data every day. By studying news classification, we may be able to help governments or relevant departments organize news more effectively or help online publishers manage their news articles more efficiently. Consequently, news classification studies are of great relevance to the real world [25].

In this study, the following tasks were performed to classify news articles:

1. A review of text classification literature inside and outside China, as well as some of the more popular approaches to deep learning, was conducted. To obtain the experimental data, a self-coded web crawler was used to obtain news articles related to sports, entertainment, and technology, which were organized into distinct datasets.

2. The data were cleaned, deduplicated, segmented, and vectorized to facilitate the construction of textual features using the BiLSTM network.

3. The hyperparameters of the LSTM network were optimized to improve the model accuracy, and a comparison between the BiLSTM and LSTM networks was conducted to examine whether the additional contextual information obtained by the BiLSTM network was relevant for text classification. In the experiment, the BiLSTM network achieved an accuracy of 94.13%.

In the era of big data, the rapid development of Internet technologies has brought considerable convenience to our daily lives but has also introduced many problems. These problems include the extraction of useful information from the enormous volumes of data, accurate (and automated) recognition, and the segmentation of new words. Although deep learning models perform well in text classification, they have not often been used for text classification. Instead, the LSTM network and its model extensions have been used for applications such as chatbots, seq2seq, machine translation, language recognition, and text generation. Nonetheless, most NLP models employ concepts or parts of the LSTM architecture, making it important to continue studying it.

## Outlook

The main focus of this study was the use of the LSTM deep learning model to classify news articles and the optimization of model parameters to improve LSTM accuracy. Although a high level of accuracy was achieved in this study, some areas require further improvement.

1. Currently, word segmentation does not perform well for new words, such as locations and names. This problem has yet to be addressed in this study.

2. Because our focus was on model optimization, no attempt was made to extend the LSTM model, for example, by incorporating an attention mechanism or convolutional core.

3. Because the experimental dataset was small, the model performed well on the training dataset but less well on the testing dataset. Mild overfitting was observed. Although dropout regularization has been used in many layers, the overfitting problem has not yet been fully resolved.

4. Although the number of texts was small, they had high word counts. Consequently, long training times were required. Furthermore, the dataset size was limited by hardware constraints. In the future, we could consider using a server to run the model and improve its efficacy.

5. Textual data were obtained using a self-coded web crawler. However, these news articles also contained non-textual data (such as images,) and the regular expressions also collected non-news textual data (such as "scan code"). As these created discontinuities in the extracted text, we should consider using open-source datasets in the future.

## Author Contributions

**Conceptualization:** Chen Liu.

**Data curation:** Chen Liu.

**Formal analysis:** Chen Liu.

**Investigation:** Chen Liu.

**Methodology:** Chen Liu.

**Project administration:** Chen Liu.

**Resources:** Chen Liu.

**Software:** Chen Liu.

**Supervision:** Chen Liu.

**Validation:** Chen Liu.

**Visualization:** Chen Liu.

**Writing – original draft:** Chen Liu.

**Writing – review & editing:** Chen Liu.

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
