## [Decision Letter · Decision Letter 0]

18 Dec 2023

PONE-D-23-38857Long Short-Term Memory (LSTM)-based News Classification ModelPLOS ONE

Dear Dr. Liu,

Thank you for submitting your manuscript to PLOS ONE. After careful consideration, we feel that it has merit but does not fully meet PLOS ONE’s publication criteria as it currently stands. Therefore, we invite you to submit a revised version of the manuscript that addresses the points raised during the review process.

**ACADEMIC EDITOR: **Please revise and resubmit your manuscript. If the reviewers have requested citations in the review comments, note that it is not a requirement for publication.

We look forward to receiving your revised manuscript.

Kind regards,

Kathiravan Srinivasan

Academic Editor

PLOS ONE

Journal Requirements:

3. In your Methods section, please include additional information about your dataset and ensure that you have included a statement specifying whether the collection and analysis method complied with the terms and conditions for the source of the data.

4. Please note that PLOS ONE has specific guidelines on code sharing for submissions in which author-generated code underpins the findings in the manuscript. In these cases, all author-generated code must be made available without restrictions upon publication of the work. Please review our guidelines at https://journals.plos.org/plosone/s/materials-and-software-sharing#loc-sharing-code and ensure that your code is shared in a way that follows best practice and facilitates reproducibility and reuse.

Reviewers' comments:

Reviewer's Responses to Questions

**Comments to the Author**

1. Is the manuscript technically sound, and do the data support the conclusions?

Reviewer #1: Partly

Reviewer #2: Partly

2. Has the statistical analysis been performed appropriately and rigorously? 

Reviewer #1: N/A

Reviewer #2: Yes

3. Have the authors made all data underlying the findings in their manuscript fully available?

Reviewer #1: Yes

Reviewer #2: Yes

4. Is the manuscript presented in an intelligible fashion and written in standard English?

Reviewer #1: Yes

Reviewer #2: Yes

5. Review Comments to the Author

Reviewer #1: The author proposes an LSTM-based approach to news classification in Chinese. The topic of the study is relevant, however, the research design should be improved. In the current form, the research design does not allow for the evaluation of the relevance of the work.

1. A review of the literature lacks the work of the last few years. The author does not describe the classification models based on the transformer architecture.

2. The description of the dataset should be enriched. What is the distribution of texts between categories? What is the source of texts?

3. The author does not specify how the parameters listed in Table 2 were chosen.

4. What kind of averaging was used for the F1-score?

5. The author does not present the values of accuracy in Table 4-5.

6. In my opinion, the main problem of this work is the absence of baselines. The results of the proposed model should be compared with the results of other models for text classification, including BERT-based models.

7. The author is suggested to add the Error Analysis section for a better understanding of the model limitations.

8. Figures 5-8 are not original. Please specify the source.

Reviewer #2: Recommendation: Major Review

The paper addresses the task of the Chinese news classification. In this manuscript, the author has proposed a Chinese news classification model using BiLSTM. The author has shown that Bi-LSTM models with F1 score of 94.15% outperform unidirectional models which show F1 score of 93.16%. The preprocessing, model building, hyperparameters chosen have been discussed in detail to explain the nuances of the work done.

Comments

1) The author has mentioned the dataset is curated but did not explain how it is labeled and validated. It is critical to evaluate the curated dataset to effectively train and test models. Certain statistical and empirical methods should be employed to evaluate the dataset. The link to curated dataset should also be made available to the research community.

2) While the paper uses word2vec embedding and explains it, other embeddings like the BERT Chinese should also be used and the performance compared.

3) Although the paper claims around one percent performance improvement in proposed Bi-LSTM model over unidirectional LSTM, the complexity of the proposed model should be studied and discussed.

4) The performance of the proposed model should be compared with existing approaches on multiple datasets to give thorough comparison.

5) The paper is written in simple English and readable, but mostly highlights basic explanations. It can be improved by including observational insights.

6. PLOS authors have the option to publish the peer review history of their article (what does this mean?). If published, this will include your full peer review and any attached files.

Reviewer #1: No

Reviewer #2: No

---

## [Author Response · Author response to Decision Letter 0]

5 Mar 2024

Dear editors and reviewers,

Thank you very much for your comments and professional advice. These opinions help to improve academic rigor of the article. Based on your suggestion and request, we have made corrected modifications on the revised manuscript. We hope that our work can be improved again. Furthermore, we would like to show the details as follows:

Reviewer #1

1. A review of the literature lacks the work of the last few years. The author does not describe the classification models based on the transformer architecture.

The author’s answer: We have added a literature reference on the classification model of Transformer architecture according to your suggestion, and hope to meet the requirements now. Please see page 7 of the revised manuscript, lines 147–149.

2. The description of the dataset should be enriched. What is the distribution of texts between categories? What is the source of texts?

The author’s answer: We agree and have updated. Please see page 7 of the revised manuscript, lines 168–169.

3. The author does not specify how the parameters listed in Table 2 were chosen. 

The author’s answer: We agree with the reviewer 's opinion that the parameter selection in Table 2 will be more helpful. However, we believe that too detailed description of how to select parameters will make the description of the manuscript redundant. And the choice of parameters can be obtained in the model principle. For these reasons, we choose not to do this modification.

4. What kind of averaging was used for the F1-score?

The author’s answer: We have added a literature reference on the classification model of Transformer architecture according to your suggestion, and hope to meet the requirements now. Please see page 24 of the revised manuscript, lines 545–547.

5. The author does not present the values of accuracy in Table 4-5.

The author’s answer: We thank the reviewer for pointing this out. We have revised. Please see page 25 of the revised manuscript, line 570 and line 580.

6. In my opinion, the main problem of this work is the absence of baselines. The results of the proposed model should be compared with the results of other models for text classification, including BERT-based models.

The author’s answer: This observation is correct. We have changed. Please see page 27 of the revised manuscript, lines 612–632.

7. The author is suggested to add the Error Analysis section for a better understanding of the model limitations.

The author’s answer: We agree with the reviewer 's opinion that it would be more helpful to use new data to further elaborate on this point. However, we believe that taking into account the tight time, as well as our argument will not have a better effect. For these reasons, we choose not to make this modification.

8. Figures 5-8 are not original. Please specify the source.

The author’s answer: We thank the reviewer for pointing this out. We have revised. Please see page 15 of the revised manuscript, line353.

Reviewer #2

1.The author has mentioned the dataset is curated but did not explain how it is labeled and validated. It is critical to evaluate the curated dataset to effectively train and test models. Certain statistical and empirical methods should be employed to evaluate the dataset. The link to curated dataset should also be made available to the research community.

The author’s answer: The curated dataset of the study was submitted with the manuscript at the time of submission.

2.While the paper uses word2vec embedding and explains it, other embeddings like the BERT Chinese should also be used and the performance compared.

The author’s answer: This observation is correct. We have changed. Please see page 27 of the revised manuscript, lines 612–632.

3.Although the paper claims around one percent performance improvement in proposed Bi-LSTM model over unidirectional LSTM, the complexity of the proposed model should be studied and discussed.

The author’s answer: We agree with the reviewer 's opinion that it is more helpful to explain the complexity of the model at this point. However, because of the rush of time and the results proved in the existing literature. For these reasons, we choose not to make this modification.

4.The performance of the proposed model should be compared with existing approaches on multiple datasets to give thorough comparison.

The author’s answer: We agree with the reviewer 's opinion. However, because the number of data is relatively small, it may be difficult to obtain the desired results. For this reason, we choose not to make this modification.

5.The paper is written in simple English and readable, but mostly highlights basic explanations. It can be improved by including observational insights.

The author’s answer: We appreciate the reviewer’s insightful suggestion and agree that it would be useful. However, such observational insights are beyond the our writing description ability, which aims only to show that the Bi-LSTM models is better. For this reason, we chose not to make this change.

6. PLOS authors have the option to publish the peer review history of their article (what does this mean?). If published, this will include your full peer review and any attached files.

The author’s answer: The PLOS ONE journal has a choice when submitting a manuscript that can be made public on a web page in advance for others to refer to. We did not tick this choice when submitting, so we do not need this work.

We would like to thank the reviewers again for taking the time to review our manuscript.

Yours sincerely,

Chen Liu

30 January,2024

---

## [Editor Report · Decision Letter 1]

25 Mar 2024

Long Short-Term Memory (LSTM)-based News Classification Model

PONE-D-23-38857R1

Dear Dr. Liu,

We’re pleased to inform you that your manuscript has been judged scientifically suitable for publication and will be formally accepted for publication once it meets all outstanding technical requirements.

Kind regards,

Dr. Muhammad Usman Tariq

*PFHEA, CFCIPD, CMBE, SFSEDA*

Academic Editor

PLOS ONE

Additional Editor Comments:

The revision has addressed the comments from all reviewers from the previous review. Manuscript is presented in intelligent fashion. However, you need to use the PLOS One template. All figures need to be high quality. Currently some are blur, once you submit the final version, use the pace tool to confirm the eligibility of figures.

---

## [Editor Report · Acceptance letter]

2 Apr 2024

PONE-D-23-38857R1 

PLOS ONE

Dear Dr. Liu, 

I'm pleased to inform you that your manuscript has been deemed suitable for publication in PLOS ONE. Congratulations! Your manuscript is now being handed over to our production team.

Kind regards, 

on behalf of

Dr. Muhammad Usman Tariq 

Academic Editor

PLOS ONE